# Research Trends of Vitamin D Metabolism Gene Polymorphisms Based on a Bibliometric Investigation

**DOI:** 10.3390/genes14010215

**Published:** 2023-01-14

**Authors:** Mohamed Abouzid, Marta Karaźniewicz-Łada, Basel Abdelazeem, James Robert Brašić

**Affiliations:** 1Department of Physical Pharmacy and Pharmacokinetics, Faculty of Pharmacy, Poznan University of Medical Sciences, Rokietnicka 3 St., 60-806 Poznan, Poland; 2Doctoral School, Poznan University of Medical Sciences, 60-812 Poznan, Poland; 3Department of Internal Medicine, McLaren Health Care, Flint, MI 48532, USA; 4Department of Internal Medicine, Michigan State University, East Lansing, MI 48823, USA; 5Section of High Resolution Brain Positron Emission Tomography Imaging, Division of Nuclear Medicine and Molecular Imaging, The Russell H. Morgan Department of Radiology and Radiological Science, The Johns Hopkins University School of Medicine, Baltimore, MD 21287, USA

**Keywords:** bibliometry, calcitriol, cancer, citation, genetic variant, genome-wide association study, index, prohormone, publication, single nucleotide polymorphism

## Abstract

Vitamin D requires activation to show its pharmacological effect. While most studies investigate the association between vitamin D and disease, only a few focus on the impact of vitamin D metabolism gene polymorphisms (vitDMGPs). This bibliometric study aims to provide an overview of current publications on vitDMGPs (*CYP27B1*, *CYP24A1*, *CYP2R1*, *CYP27A1*, *CYP2R1*, *DHCR7/NADSYN1*), compare them across countries, affiliations, and journals, and inspect keywords, co-citations, and citation bursts to identify trends in this research field. CiteSpace^©^ (version 6.1.R3, Chaomei Chen), Bibliometrix^©^ (R version 4.1.3 library, K-Synth Srl, University of Naples Federico II, Naples, Italy), VOSviewer^©^ (version 1.6.1, Nees Jan van Eck and Ludo Waltman, Leiden University, Leiden, Netherlands) and Microsoft^®^ Excel 365 (Microsoft, Redmond, Washington, USA) classified and summarized Web of Science articles from 1998 to November 2022. We analyzed 2496 articles and built a timeline of co-citations and a bibliometric keywords co-occurrence map. The annual growth rate of vitDMGPs publications was 18.68%, and their relative research interest and published papers were increasing. The United States of America leads vitDMGPs research. The University of California System attained the highest quality of vitDMGPs research, followed by the American National Institutes of Health and Harvard University. The three productive journals on vitDMGPs papers are *J. Steroid. Biochem. Mol. Biol., PLOS ONE*, and *J. Clin. Endocrinol. Metab*. We highlighted that the vitDMGPs domain is relatively new, and many novel research opportunities are available, especially those related to studying single nucleotide polymorphisms or markers in a specific gene in the vitamin D metabolism cycle and their association with disease. Genome-wide association studies, genetic variants of vitDMGPs, and vitamin D and its role in cancer risk were the most popular studies. *CYP24A1* and *CYB27A1* were the most-studied genes in vitDMGPs. Insulin was the longest-trending studied hormone associated with vitDMGPs. Trending topics in this field relate to bile acid metabolism, transcriptome and gene expression, biomarkers, single nucleotide polymorphism, and fibroblast growth factor 23. We also expect an increase in original research papers investigating the association between vitDMGPs and coronavirus disease 2019, hypercalcemia, Smith–Lemli–Opitz syndrome, 27-hydroxycholesterol, and mendelian randomization. These findings will provide the foundations for innovations in the diagnosis and treatment of a vast spectrum of conditions.

## 1. Introduction

Vitamin D is a fat-soluble vitamin obtained mainly from sun exposure or alternatively from dietary sources and supplements [1,2,3]. Vitamin D is a prohormone that requires activation before exhibiting its pharmacological actions [4]. Therefore, it undergoes metabolism in the liver by *CYP2R1* and *CYP27A1* to yield 25-hydroxyvitamin D (25(OH)D) [5]. Further metabolism in the kidney by *CYP27B1* yields calcitriol (1, 25(OH)_2_D_3_), the active form of vitamin D [6,7]. Calcitriol binds to the vitamin D receptor (*VDR*), dimerizes with the retinoid receptor, and translocates to the nucleus to bind to the vitamin D response element. This complex can then induce or repress gene transcription depending on the target gene, either co-activators or co-repressors [8,9]. Therefore, vitamin D-related disorders may be caused by polymorphic variants of genes coding protein molecules involved in vitamin D metabolism and transport [10].

Despite the complicated pathway of vitamin D metabolism, vitamin D and *VDR* remain the “superstars” and the central points of investigation. For example, in the cardiovascular field, several observational and preliminary studies have shown the influence of seasonal patterns on vitamin D levels and the inverse association between serum 25(OH)D levels and the risk of cardiovascular disease events [11,12,13,14,15]. Moreover, vitamin D levels have been connected with an elevated risk of myocardial infarction, stroke, cardiovascular disease mortality, and heart failure in case-control and other prospective epidemiologic studies [16,17,18]. However, a meta-analysis of 21 randomized controlled trials showed no benefits of vitamin D supplementation for CVD [19]. Moreover, recent studies have highlighted the importance of analyzing vitamin D serum levels and its metabolites in multiple diseases [20,21]. The exact mechanism of how a deficiency in vitamin D can have an extended influence on several organs requires more elucidation.

The bibliometric investigation is a robust methodology that quantitatively uses the literature metrics or indicators to measure the research performance in a specific field. The method assists in identifying novel and emerging research areas and is widely used in medical research [22]. Through bibliometric analysis, researchers can quickly and comprehensively achieve influential articles, main research fields, and new research directions [23]. Compared with traditional reviews, bibliometric-based analyses can provide a more comprehensive perspective on research trends, making the data more objective [23]. With the development of bibliometrics, dozens of software tools for conducting bibliometric analysis are available for researchers [24].

Recently, there have only been a few bibliometric studies investigating the trend research of *VDR* [25] and vitamin D [26,27], and the association of vitamin D with reproductive health [28], bone metabolism [29], multiple sclerosis [30], and coronavirus disease 2019 [31]. However, no published bibliometric analysis covered any vitamin D metabolism gene polymorphisms (henceforth referred to as vitDMGPs for simplicity). It is also worth mentioning that, generally, there is a lack of review articles investigating vitDMGPs [32,33]. Hence, comprehensive knowledge of the global research status, current hotspots, and future trends of vitDMGPs remains mainly unknown.

Therefore, this study systematically performed a bibliometric-based analysis to evaluate vitDMGPs studies from 1950 to 2022. By taking advantage of CiteSpace^©^ (version 6.1.R3, Chaomei Chen) [34], VOSviewer^©^ (version 1.6.1, Nees Jan van Eck and Ludo Waltman, Leiden University, Leiden, Netherlands) [35], and Bibliometrix^©^ (R version 4.1.3 library, K-Synth Srl, University of Naples Federico II, Naples, Italy) [36], we comprehensively analyzed the publication output, disciplinary composition, countries/regions, institutions, authors, journals, top-cited articles, references, and appeared keywords. Our results draw a visualization map of the global research landscape of vitDMGPs, helping researchers, especially those new to this field, to deepen their understanding of the current status and future research directions.

## 2. Materials and Methods

### 2.1. Database and Search Methodology

We primarily used the Web of Science (WOS) subscription from Clarivate Analytics to search the literature on vitamin D metabolism polymorphisms in October 2022. The selected WOS database included *CCR-EXPANDED, CPCI-SSH, CPCI-S, A&HCI, BKCI-SSH, SCI-EXPANDED, SSCI, BKCI-S,* and *ESCI.* The following adjustments were made: document type → “article”; language → “English”; and period → from “1950” to “2022”. Field tag “TS = Topic,” was used in the advanced search field, which expanded the lookup for the query words in the title, abstract, and keywords. We aimed to investigate vitDMGPs previously identified in our previous paper (*CYP27B1, CYP24A1, CYP2R1, CYP27A1, CYP2R1, DHCR7/NADSYN1*) [32]; therefore, we deployed the following query:TS = (*CYP27B1 OR CYP24A1 OR CYP2R1 OR CYP27A1 OR DHCR7 OR NADSYN1*)

### 2.2. Data Extraction

Data extraction was performed in a single sitting to ensure the accuracy of the data (articles can be added retrospectively in WOS). Data (record content = full record and cited references) were obtained from the WOS database as separate files with different types to meet the minimum criteria for each analysis software (VOSviewer^©^ (version 1.6.1, Nees Jan van Eck and Ludo Waltman, Leiden University, Leiden, Netherlands) [35], “tab delimited”, *N* = 5 files; Bibliometrix^©^ (version 1.6.1, Nees Jan van Eck and Ludo Waltman, Leiden University, Leiden, Netherlands) [35] and CiteSpace^©^ (version 6.1.R3, Chaomei Chen) [34], “plain text file”, *N* = 5 files). For each type, the files were combined into one file. No duplicates were detected. Microsoft Excel (provided via Microsoft^®^ 365 academic license) was used to record publication information: number, year, country, organization, journal, author, H-index, journal impact factor, and total times cited. The following formula was used to calculate the relative research interest (*RRI*) [37]:$$RRI=\frac{Number\;of\;vitDMGPs\;publications\;per\;year}{Number\;of\;all\;publications\;\left(all\;disciplines\right)\;per\;year}\;$$

### 2.3. Data Analysis

We used Microsoft^®^ Excel 365 (Microsoft, Redmond, Washington, USA) and Bibliometrix^©^ (R version 4.1.3 library) [36] to report frequencies of publication, country, organization, journal, author, H-index, impact factor, and total times cited. VOSviewer^©^ (version 1.6.1) [35] analyzed and visualized the co-occurrence of keywords. Bibliometrix^©^ was used to report frequencies of the annual most trending three keywords since 2001. The co-citation of references timeline was created using CiteSpace^©^ (version 6.1.R3) [34] for a comprehensive analysis from January 2006 to December 2022, using k = 25 for the scaling factor (*N* = 2278 qualified records). Centrality was calculated for 984 nodes. We performed all work on a 32 GB desktop computer equipped with AMD Ryzen^TM^ 7 3700X 8-Core Processor and GeForce GTX^TM^ 1070 graphic card.

## 3. Results

### 3.1. Publications about Vitamin D Metabolism Gene Polymorphisms (vitDMGPs)

WOS generated an initial 3276 publications that were extracted and filtrated accordingly. Duplicate publications and those with missing records, including missing citing references or publication years, or those with anonymous authors/references, were eliminated (*N* = 16). Moreover, we included only original published articles and review articles, and we excluded meeting abstracts, proceeding papers, book chapters, editorials, letters, early access articles, corrections, retractions, data papers, and news items (*N* = 764). Thus, a total of 2496 papers were selected for further analysis. The number of citations of all papers associated with vitDMGPs was 37,519 (35,420 not including self-citations). The H-index was 109. The average citation number for entire articles was 26.95 times. The citation details (total citations, total annual citations, and normalized total citations (the actual count of citing items divided by the expected citation rate for documents with the same document type, year of publication, and subject area)) of the top 10 cited articles about vitDMGPs are shown in Table 1. Normalized total citations (field-weighted citation impact) can be interpreted as follows: “Exactly 1.00 means that the output performs just as expected for the global average. More than 1.00 means that the output is more cited than expected according to the global average; for example, 1.48 means 48% more cited than expected. Less than 1 means that the output is cited less than expected according to the global average; for example, 0.91 means 9% less cited than expected” ([38], p.10). When a document was assigned to more than one subject area, an average of the ratios of the actual to expected citations was used.

### 3.2. Synopsis of Publications about Vitamin D Metabolism Gene Polymorphisms (vitDMGPs)

The papers’ timespan was from 1998 to 2022, and the annual growth rate was 18.68%. The most productive year was 2021, with 243 publications, followed by 2020 and 2019, with 209 and 205 papers, respectively (Figure 1a).

### 3.3. Countries of Publications and Scientific Categories

The United States of America (USA) published most papers (938; 37.58%), followed by China (411; 16.47%). Canada published 194 papers, accounting for 7.77% (Figure 1b). Research from England has the highest average number of citations (45.56 times), followed by the Netherlands (45.34 times) and Sweden (43.66 times). The USA, England, Canada, and Germany had the highest H-index values, with 89, 44, 43, and 43, respectively (Figure 1c). The top three scientific disciplines were endocrinology metabolism (20%), biochemistry, molecular biology (19%), and genetics heredity (11%), as seen in Figure 1d.

### 3.4. Journals of Publications and Scientific Disciplines

*J. Steroid Biochem. Mol. Biol.* (103 publications, impact factor for 2021 (IF) 5.011, CiteScore 8.8) published the most papers on vitDMGPs, followed by PLOS *ONE* (83 publications, IF 3.752, CiteScore 5.6) and *J. Clin. Endocrinol. Metab.* (48 publications, IF 6.134, CiteScore 9.1) (Figure 2a). The top 3 productive journals contributed to publishing 9.4% of vitDMGPs papers. Out of 863 journals, the top 10 journals published 12 or more papers on vitDMGPs, accounting for 20.3% of all papers related to vitDMGPs.

### 3.5. Authors of Publications

Out of 14,046 authors who published papers associated with vitDMGPs, the top 10 authors account for 19% of the total vitDMGPs publications (Figure 2b). The first contributing author to vitDMGPs research was Sakaki T. He co-authored 61 publications and had an H-index of 24. He was followed by Jones G (41 publications, H-index 22) and Porter FD and Tuckey RC (33 publications, H-index 21 and 18, respectively).

### 3.6. Affiliations of Publications

The most productive organization on vitDMGPs research is the American National Institutes of Health, with 99 publications, followed by the University of California System, Harvard University, McGill University, and Harvard Medical School, with 95, 83, 71, and 64 publications, respectively (Figure 2c).

The citation frequency of papers from the National Institutes of Health ranked first (5382 times), followed by Harvard University (5173 times) and the University of California System (4664 times). The H-index of the University of California System ranked first (37), followed by Harvard University (35) and the National Institutes of Health (34). We also analyzed the association between countries, affiliations, and shared keywords, showing the contribution and impact of each institution toward a particular specialty (Figure 3).

### 3.7. Analysis of Co-Cited References

The timeline view of clustered maps of 2321 co-cited references from 2006 to 2022 yielded 19 clusters, according to keywords. According to Chen, “the modularity of a network measures the extent to which a network can be decomposed to multiple components, or modules” ([49], p.82). Therefore, a modularity Q of 0.7696 is relatively high, so the network is reasonably divided into loosely coupled clusters. The Silhouette value can be used as an indicator for the quality of clustering configuration, and it ranges between −1 and 1 [49]. The mean Silhouette score of 0.8909 suggests relatively high homogeneity of the clusters [49]. As shown in Figure 4, the newest clusters appear to have lighter colors, such as the 9th, 12th, and 13th, and they focus on (“COVID-19”, Coronavirus disease 2019), (“smith–”, Smith Lemli Opitz Syndrome), and (“7-dhc”, 7-Dehydrocholesterol Reductase), respectively. Other topics seem to have a good trend after 2010, including those relating to hypercalcemia, CYP24A1, and mendelian randomization.

Clusters were then ranked by Silhouette score. A higher Silhouette score indicates a higher homogeneity (maximum = 1). Cluster labels are shown in Table 1 according to different labels: latent semantic indexing (LSI), Log-likelihood ratio (LLR), and mutual information (MI) for better clarity of the dynamicity of the publications on the timeline (Table 2). LSI shows the most important aspect of a cluster, whereas LLR and MI tend to reflect a unique aspect of it. Moreover, analysis of the strongest citation bursts shows the top 25 references that impacted science from 2006 to 2021 [1,5,8,40,41,45,52,55,56,57,58,59,60,61,62,63,64,65,66,67,68,69,70,71,72] (Figure 5).

### 3.8. Hotspots of Papers Related to Vitamin D Metabolism Gene Polymorphisms (vitDMGPs)

Hotspot topics were identified by designing the authors’ keywords’ map and selecting the keywords mentioned at least 10 times. Of 4405 keywords, 114 were chosen and classified into 6 clusters (Table 3). The most frequently studied genes were *CYP24A1, CYP27B1,* and *CYP27A1*, with frequencies of 192, 146, and 89, respectively. Smith–Lemli–Opitz Syndrome, cerebrotendinous xanthomatosis, and breast cancer were the most frequent disease keywords with 67, 59, and 42 times, respectively. The most frequent keywords plus were expression, metabolism, risk, 1,25 dihydroxyvitamin D_3_, vitamin D, and D deficiency, with 475, 278, 256, 229, 213, and 193 times, respectively (Figure 6). The overlay visualization shows the average publication year for each paper (Figure 7). The list of the annual trending top three keywords since 2001 is shown in Figure 8.

## 4. Discussion

### 4.1. Trends of Publications Related Vitamin D Metabolism Gene Polymorphisms (vitDMGPs)

Since 1998, the highest record of published papers and RRI for vitDMGPs was in 2021. Still, the research amount generated is considered relatively low compared to our previous VDR analysis indicating more upcoming novel ideas associated with vitDMGPs [25]. The USA had the most impact on this research sector because it had the highest H-index. China was in second place after the USA in paper amount but had a lower H-index than Canada (ranked 3^rd^ in publication amount) and comparable H-index with Japan and Australia, which ranked 4^th^ and 7^th^ in paper amount. However, due to our comprehensive analysis, we cannot wholly associate H-index with quality. Looking at Figure 4, we found that China had participated less in popular research topics in the fields of vitamin D, *CYP24A1*, *CYP27B1*, VDR, and *CYP27A1*. The reason was lower Chinese production on these publications over time since China only reached 2nd place of publication production on vitDMGPs in 2017 with 423 publications. Hence, it requires more time to see the impact. Analyzing countries’ production over time shows that countries such as Japan, Canada, and Germany had a steady increase in their production over the years, while the USA surpassed other countries in this field in 1998 and the gap has increased notably since 2004. It is no wonder that among the top 10 productive affiliations, seven are American. Generally, the collaboration between the USA is expanded to Canada, Australia, Japan, China, and European countries (Figure 3).

Concerning the recent trending keywords, first, we notice in Figure 8 that key words remain in trend from 2007 until 2021. This more extended period indicates the expansion of the topic on the horizontal level to investigate several elements at once. Hence, it is essential while investigating the most trending words to look at Figure 7 to see the average publication range for each keyword. For example, “insulin” was the longest-trending studied hormone from 2001 until 2019; however, its average publication year was 2010, indicating it is an outdated topic. We did not observe this word in the co-occurrence map when we increased the frequency to 10 times instead of 5. On the other side, the term “vitamin D” was trending from 2014 to 2020 even though we did not use “vitamin D” in our query, which indicates the robust relationship between vitDMGPs and vitamin D in the bibliometric analysis. Vitamin D also had a relatively high average publication year, and it became the most frequently used author keyword.

New topics relate to mendelian randomization, hypercalcemia, lipid metabolism, mineral metabolism, single nucleotide polymorphism/SNPs, bile acid metabolism, and transcriptome. It is also essential to relate to the co-cited references because it might provide a new area to study. In Figure 4, “COVID-19”, “27-hc/27-hydroxycholesterol”, and “smith/Smith–Lemli–Opitz” clusters will have a new appearance indicating the possibility of investigating these domains in the future. Despite the outdated topic of Smith–Lemli–Opitz as a keyword (Figure 7), its occurrence in the co-citation map may also show that some authors are trying to rediscover or restudy old topics. However, there is no guarantee that these will continue and they might be replaced with other topics, such as hypercalcemia and *CYP24A1*, since these appeared to be co-cited clusters and also appeared in trending topics. Generally speaking, *CYP24A1*, *CYP27B1,* and *CYP27A1* were the most studied genes, so upcoming research may either continue the trend and examine the SNPs of these CYPs or inspect other genes that are not trending and have a low appearance on vitDMGPs domain. However, the latter option might have a higher risk. Surprisingly, we noticed that the association between vitDMGPs and cardiovascular risk is poorly studied (Figure 4) despite the ongoing scientific battle between vitamin D and cardiovascular disease.

### 4.2. Studies Focused on Vitamin D Metabolism Gene Polymorphisms (vitDMGPs)

#### 4.2.1. Global Most Cited Documents

In our analysis, the three most globally cited documents were those published by Salzman et al. [39], Wang et al. [40], and Ahn et al. [41], with 1299, 1158, and 579 citations, respectively. The paper “Cell-Type Specific Features of Circular RNA Expression” by Salzman et al. [39], published in 2013, has gained too much scientific attention since the authors have tested eight RNAs, ABTB1, FAT1, HIPK3, CYP24A1, LINC00340, LPAR1, and PVT1 to conclude that the repertoire of genes expressing circular RNA, the ratio of circular to linear transcripts for each gene, and even the pattern of splice isoforms of circular RNAs from each gene were cell-type specific—we focus on reporting findings of CYP24A1 as vitDMGPs. The authors reported that the CYP24A1 gene was the most highly expressed circular RNA identified in a lung cancer cell line, A549. They linked this expression to the pathogenesis of many carcinomas [39].

The 2^nd^ and 3^rd^ cited papers were genome-wide association studies (GWAS). The paper “Common genetic determinants of vitamin D insufficiency: a genome-wide association study” by Wang et al. [40] published in 2010 was a meta-analysis in which they analyzed 25-hydroxyvitamin D concentrations in 33,996 individuals. The study concluded that genetic variation at several loci—*rs2282679* in *GC*, *rs12785878* near *DHCR7*, *rs10741657* in *CYP2R1*, and *rs6013897* in *CYP24A1*—have a noticeably elevated risk of vitamin D insufficiency. In the third paper, titled “Genome-wide association study of circulating vitamin D levels” by Ahn et al. [41], published in 2010, they analyzed 25(OH)D concentrations in 4501 individuals and found that *NADSYN1/DHCR7* and *CYP2R1* had strong genome-wide significant associations with 25(OH)D.

#### 4.2.2. Burst Documents That Co-Cited Vitamin D Metabolism Gene Polymorphisms (vitDMGPs)

The studies by Bray et al. [72], Jiang et al. [71], and Christakos et al. [5] had the most citation bursts since 2016. The study “Global cancer statistics 2018: GLOBOCAN estimates of incidence and mortality worldwide for 36 cancers in 185 countries” by Bray et al. [72] has analyzed 36 cancers’ incidence and mortality in 20 regions and estimated 18.1 million new cancer cases and 9.6 million cancer deaths. Although vitDMGPs are not mentioned in the paper, having “breast cancer” as the most trending disease since 2015 associated with vitDMGPs makes it an excellent source to cite, especially since 5% to 10% of breast cancer cases are due to hereditary and genetic factors. The GWAS study by Jiang et al., titled “Genome-wide association study in 79,366 European-ancestry individuals informs the genetic architecture of 25-hydroxyvitamin D levels”, was published in 2018. In this study, the authors confirmed the association of loci *GC*, *NADSYN1/DHCR7*, *CYP2R1*, *CYP24A1*, *SEC23A,* and *AMDHD1* with 25-hydroxyvitamin D levels. It was found that 25-hydroxyvitamin D has a modest overall heritability due to common genome-wide SNPs of 7.5%, and an appreciable proportion of this total (2.84% out of 7.5%, i.e., 38%) could be explained by known genetic regions identified through GWAS [71]. The third paper is a review titled “Vitamin D: Metabolism, Molecular Mechanism of Action, and Pleiotropic Effects” by Christakos et al. [5] and was published in 2015. The authors created a well-constructed and updated review on vitamin D and, most importantly, reported its pleiotropic effect on cancer, the cardiovascular system, and the immune system.

### 4.3. Strengths and Limitations

We used three tools to validate/confirm our results and have an in-depth analysis using different features. These results can guide new researchers or those looking for upcoming trends. The nature of the timely based bibliometric study is a limitation because this information changes over time. We included only published articles, and the exclusion criteria for analyzing trending keywords were set to a minimum of five. Thus, at the time of writing this paper, newly published remarkable documents (processing papers, letters, and early access articles) were excluded from the trend analysis. We also used H-index as an indicator for the quality/novelty of the paper, but this might not be entirely accurate since there are many good papers or preliminary studies that still do not have many citations.

## 5. Conclusions

Research on vitDMGPs is relatively new compared to *VDR*, and its relative research interest is increasing. Popular studies in the field include genome-wide association studies, genetic variants of vitDMGPs, and vitamin D. *CYP24A1* and *CYB27A1* were the most explored genes, suggesting more research opportunities for other genes. Unexpectedly, the association between vitDMGPs and cardiovascular diseases is outdated, suggesting further studies covering this point or updating its recent knowledge. The trending topics for the past two years were bile acid metabolism, transcriptome and gene expression, biomarkers, single nucleotide polymorphism, and fibroblast growth factor 23. It is expected that an increase in studies investigating vitDMGPs and coronavirus disease 2019, hypercalcemia, Smith–Lemli–Opitz syndrome, 27-hydroxycholesterol, and mendelian randomization will be noticed. The results of this protocol provide the foundations for novel investigations of these conditions and a vast spectrum of other situations.

## Figures and Tables

**Figure 1 genes-14-00215-f001:**
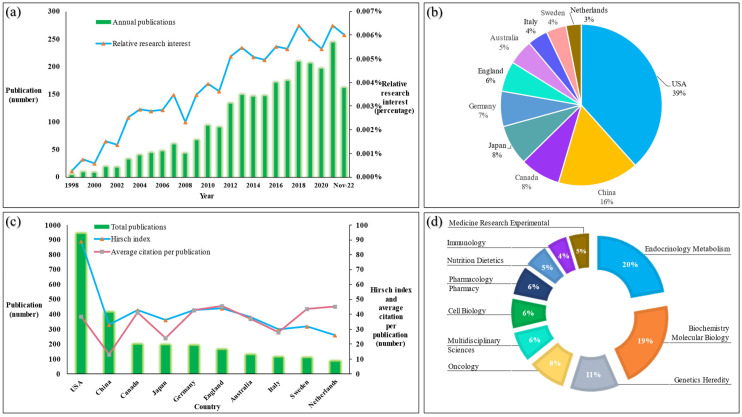
Characteristics of publications about vitamin D metabolism gene polymorphisms (vitDMGPs). (**a**) Relative research interest and annual publication number; (**b**) Global distribution of the publications among different countries; (**c**) Top 10 productive countries, their H-index, and average citation per publication; (**d**) Major (top 10) Web of Science sectors concerning vitDMGPs publications.

**Figure 2 genes-14-00215-f002:**
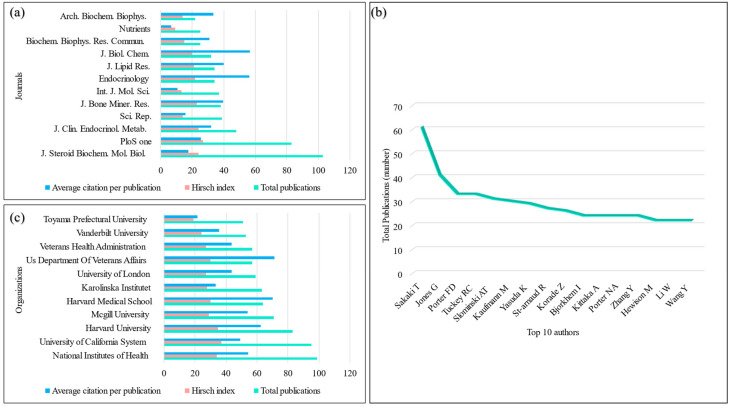
Average citations per publication, H-index, and total publications for top productive (**a**) journals and (**c**) affiliations; (**b**) Top 10 productive authors in vitDMGPs research.

**Figure 3 genes-14-00215-f003:**
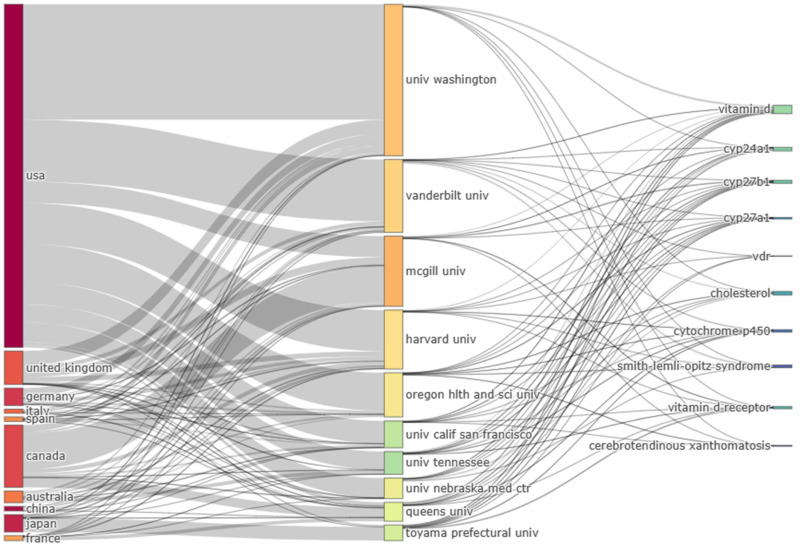
Three-field plot (*N* = 10 item/plot) of highly impacted countries (**left**), affiliations (**middle**), and keywords (**right**).

**Figure 4 genes-14-00215-f004:**
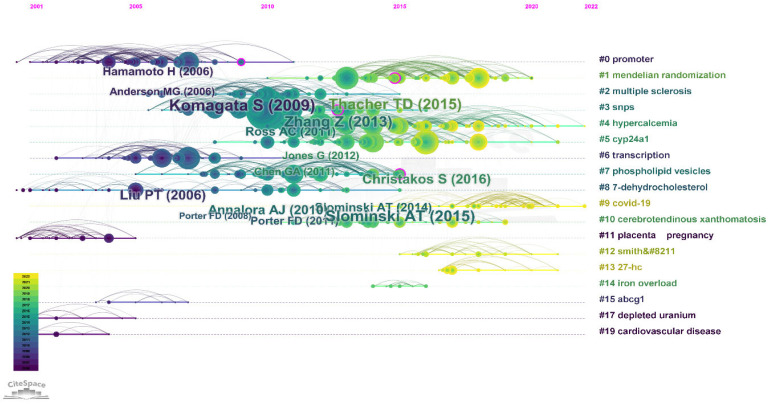
Timeline analysis co-citation network showing 19 clusters of changes in authors’ keywords over time, lighter colors indicate recent co-citation relationship. The topic of each cluster is shown on the right side of the image. A larger nodes label indicates higher centrality, starting from Komagata S (2009, cluster #0 [50]) and Slominski AT (2015, cluster #7 [51]) with 0.11 centrality, and Liu PT (2006, cluster #6 [52]), Thacher TD (2015, cluster #1 [53]) and Zhang Z (2013, cluster #3 [54]) with 0.1 centrality (marked with pink circle).

**Figure 5 genes-14-00215-f005:**
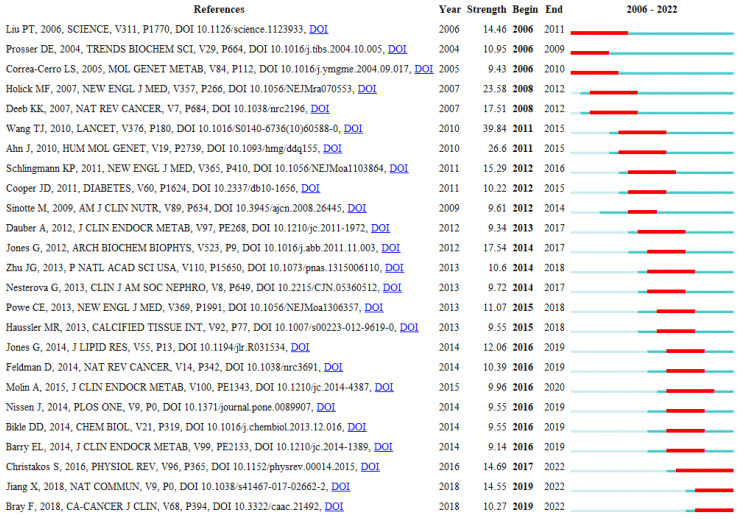
Top 25 references with the strongest citation bursts from 2006 to 2021. Liu, P.T., 2006 [52]; Prosser, D.E., 2004 [55]; Correra-Cerro, L.S., 2005 [56]; Holick, M.F., 2007 [57]; Deeb, K.K., 2007 [58]; Wang, T.J., 2010 [40]; Ahn, J., 2010 [41]; Schlingmann, K.P., 2011 [45]; Cooper, J.D., 2011 [59]; Sinotte, M., 2009 [60]; Dauber, A., 2012 [61]; Jones, G., 2012 [62]; Zhu, J.G., 2013 [63], Nesterova, G., 2013 [64]; Powe, C.E., 2013 [65]; Haussler, M.R., 2013 [8]; Jones, G., 2014 [66]; Feldman, D., 2014 [67]; Molin, A., 2015 [68]; Nissen, J., 2014 [69]; Bikle, D.D., 2014 [1]; Barry, E.L., 2014 [70]; Christakos, S., 2016 [5]; Jiang, X., 2018 [71]; Bray, F., 2018 [72].

**Figure 6 genes-14-00215-f006:**
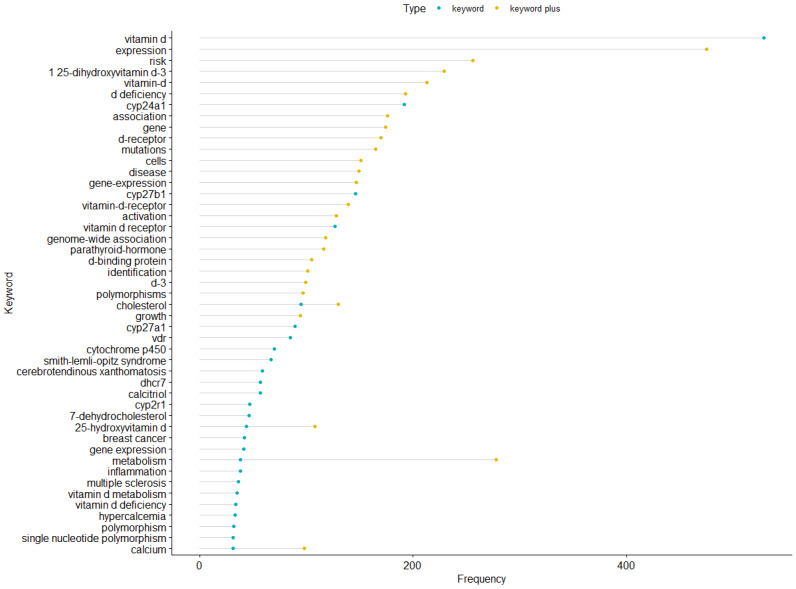
Top frequent 25 keywords and keywords plus.

**Figure 7 genes-14-00215-f007:**
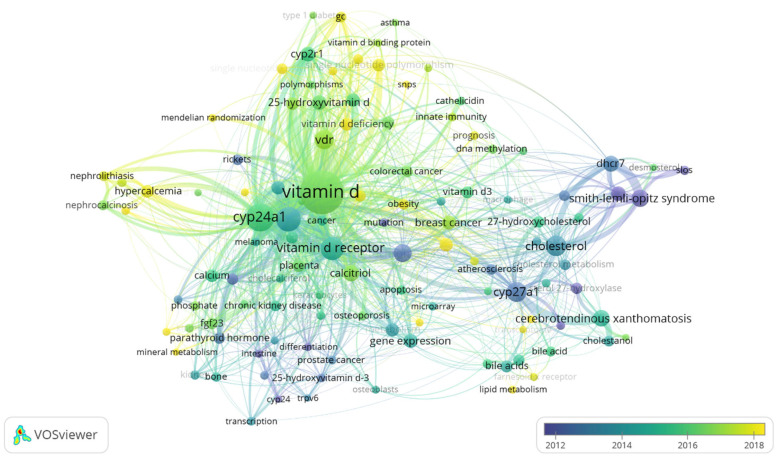
Overlay visualization of 114 most frequent author keywords. The average publication year for papers ranges from 2008 to 2019.

**Figure 8 genes-14-00215-f008:**
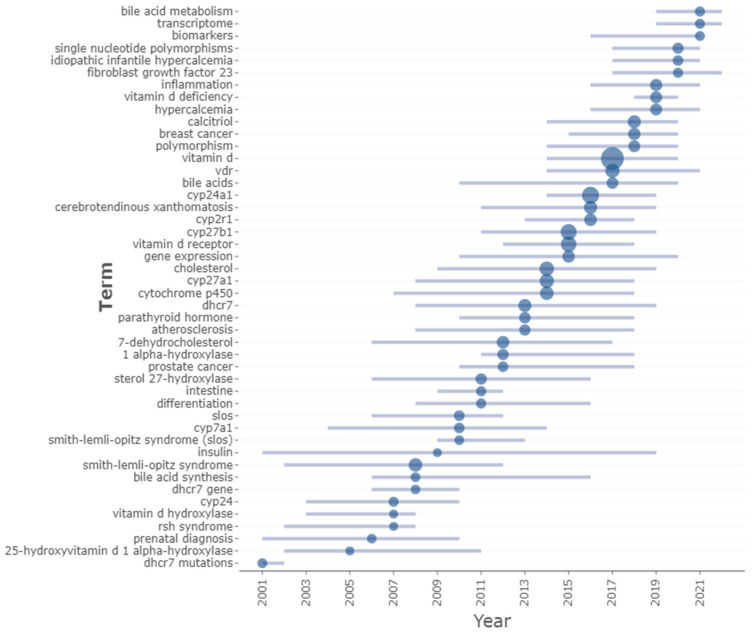
The list of the annual top three trending keywords from 2001 to 2022.

**Table 1 genes-14-00215-t001:** Most globally cited documents about vitamin D metabolism gene polymorphisms (vitDMGPs) [39,40,41,42,43,44,45,46,47,48].

Paper	DOI	TC	TC Per Year	Normalized TC
[39] Salzman, J. et al., 2013, *PLOS Genet*.	10.1371/journal.pgen.1003777	1299	129.90	27.38
[40] Wang, T.J. et al., 2010, *Lancet Lond. Engl.*	10.1016/S0140-6736(10)60588-0	1158	89.08	14.87
[41] Ahn, J. et al., 2010, *Hum. Mol. Genet.*	10.1093/hmg/ddq155	579	44.54	7.43
[42] Nelson, E.R. et al., 2013, *Science*	10.1126/science.1241908	499	49.90	10.52
[43] Schauber, J. et al., 2007, *J. Clin. Invest.*	10.1172/JCI30142	498	31.13	8.67
[44] Cheng, J.B. et al. 2004, *Proc. Natl. Acad. Sci. U.S.A.*	10.1073/pnas.0402490101	477	25.11	8.54
[45] Schlingmann, K.P. et al., 2011, *New Engl. J. Med.*	10.1056/NEJMoa1103864	382	31.83	8.02
[46] Hewison, M., 2010, *Endocrin. Metab. Clin. North Am.*	10.1016/j.ecl.2010.02.010	365	28.08	4.69
[47] Motola, D.L. et al., 2006, *Cell*	10.1016/j.cell.2006.01.037	352	20.71	6.66
[48] Petta, S. et al., 2010, *Hepatology*	10.1002/hep.23489	333	25.62	4.28

DOI—document object identifier; TC—total citations.

**Table 2 genes-14-00215-t002:** Summary of the largest 19 clusters of vitDMGPs articles with top terms mentioned (latent semantic indexing (LSI), Log-likelihood ratio (LLR), and mutual information (MI)).

ClusterID	Size	Silhouette	Top Terms (LSI) ^α^	Top Terms (LLR, p-Level) ^β^	Top Terms (MI) ^γ^	Average Year
4	63	0.922	idiopathic infantile hypercalcemia	idiopathic infantile hypercalcemia (1176.6, 10^−4^)	α-oh derivative (1.2)	2015
3	62	0.894	genetic variant	african american (576.35, 10^−4^)	antenatal cholecalciferol supplementation (1.52)	2010
1	53	0.862	controlling vitamin	mendelian randomization study (538.14, 10^−4^)	α-oh derivative (1.24)	2015
5	51	0.796	colorectal cancer	gene polymorphism (426.57, 10^−4^)	upregulated cyp24a1 (1.94)	2014
0	45	0.816	25-dihydroxyvitamin d-3	mutational analysis (339.68, 10^−4^)	α-oh derivative (0.35)	2004
2	45	0.836	multiple sclerosis	multiple sclerosis (1174.9, 10^−4^)	altered vitamin (0.73)	2009
7	42	0.956	20-hydroxyvitamin d3	20-hydroxyvitamin d3 (562.32, 10^−4^)	α-oh derivative (0.19)	2011
8	33	0.982	lemli-opitz syndrome	lemli-opitz syndrome (895.15, 10^−4^)	dhcr7-het mouse model (0.08)	2006
6	30	0.894	old theme	old theme (339.83, 10^−4^)	α-oh derivative (0.27)	2005
10	24	0.992	cerebrotendinous xanthomatosis	cerebrotendinous xanthomatosis (1407.51, 10^−4^)	early identification (0.1)	2014
9	20	0.952	cyp11a1-derived vitamin	cyp11a1-derived vitamin (119.56, 10^−4^)	vdr-associated lncrna (0.03)	2018
12	13	0.988	glial cholesterol synthesis	prescription medication (97.39, 10^−4^)	targeting 7-dehydrocholesterol reductase (0.01)	2017
11	12	0.999	intestinal caco-2 cell	thp-1 macrophage (67.29, 10^−4^)	multiple sclerosis (0.01)	2002
13	8	0.998	cancer development	emerging role (78.89, 10^−4^)		2017
14	6	0.997	deferasirox pharmacokinetics	deferasirox pharmacokinetics (77.92, 10^−4^)		2015
15	6	1	amyloid-β peptide production	amyloid-β peptide production (49.27, 10^−4^)		2005
17	5	0.998	uranium	rat (47.47, 10^−4^)		2002
19	4	0.999	cardiovascular disease	role (32.35, 10^−4^)		2002

^α^ LSI is based on a singular value decomposition of the term by document matrix [73]. ^β^ LLR uses the asymptotic distribution of the generalized likelihood ratio, which improves the statistical results. For textual analysis, LLR can be applied to smaller text volumes and allows the comparison between the significance of the occurrence of both rare and common phenomenon [74]. ^γ^ MI measures the degree of relatedness between two variables [75].

**Table 3 genes-14-00215-t003:** The main vitDMGPs research topics according to keywords.

Cluster No.	Keywords’ Counts	Focus
1	31	CYP27A1 and bile acid and lipid metabolism
2	29	CYP27B1 and vitamin D receptor and vitamin D metabolism
3	24	Vitamin D genetics and deficiency
4	13	Cytochrome P450, DHCR7, cholesterol, and Smith–Lemli–Opitz Syndrome
5	11	CYP24A1, hypercalcemia, vitamin D metabolism, and nephrocalcinosis
6	6	25(OH)D_3_ and pregnancy

## Data Availability

The raw data used for CiteSpace^©^, Bibliometrix^©^, and VOSviewer^©^ to create the bibliometric map and cluster timeline can be downloaded from Harvard Dataverse (https://doi.org/10.7910/DVN/BQGSYI) [76].

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
