# Peer review of "Research Trends of Vitamin D Metabolism Gene Polymorphisms Based on a Bibliometric Investigation"

_genes, 2023, doi:10.3390/genes14010215_

Round 1

Reviewer 1 Report

Abouzid et al in the article titled ` Novel Insights in Vitamin D Metabolism Gene Polymorphisms to Advance the Diagnosis and Treatment of a Spectrum of Diseases` describes a very novel and original way of summarizing the data in the field of Vitamin D describing so far less studied aspects, such as impact of Vitamin D metabolism gene polymorphisms (vitDMGPs).

The authors compared and analyzed 2,496 relative scientific articles regarding the countries, affiliations, journals, impact factors, citations, co-citations, using CiteSpace, Biblio-24 metrix, VOSvievew and Microsoft excel 365 to classify the articles in period from 1998 to 2022.

This can present a very powerful tool for the researchers in this field, especially the young ones, to start analyzing the relevant aspects but less studied.

The methods are very detailedly described and the quality of the figures and tables are of high quality.

The limitation ,as also the authors mention, of this study is definitely the fluid nature of publishing new articles on the similar topic that cannot be all presented in this article, but this still gives an important asset that articles like this should be more published with novel analyses of relevant data in this field in order to keep the data updated.

Author Response

We thank the reviewer for their thoughtful comments.

Reviewer 2 Report

General comments:

Vitamin D-related disorders may be caused by polymorphic variants of genes coding protein molecules involved in vitamin D metabolism and transport. However, no published bibliometric analysis covered any vitamin D metabolism gene polymorphisms (vitDMGPs). This manuscript presents a bibliometric investigation for vitDMGPs. Although this paper is interesting, I have several concerns:

1.     This paper uses bibliometric investigation to study vitDMGPs, which has certain research significance, but it is not appropriate to say that novel insights, and this paper has a limited role in promoting the diagnosis and treatment of a spectrum of diseases. Thus, The consistency of the text topic needs to be strengthened, such asResearch trends of vitamin D metabolism gene polymorphisms based on a bibliometric investigation.

2.      The time range of literature research needs to be checked. In the abstract, the research period is from 1998 to 2022, but in the Database and search methodology section, the research period is set as 1950 to 2022.

3.      In bibliometric investigation, the depth and breadth of literature research are very important, but I noticed that the scope of research in this paper is limited to articles in English, and the author should expand the scope of research in papers in other languages. In addition, processing papers, letters and early access articles may usually include the latest research results in this field, which is inappropriate to delete.

4.      In Section 3.7, the article uses some indicators to display the research results, such as Modularity Q, late semantic indexing (LSI), Log like food ratio (LLR), and mutual information (MI), but does not clearly explain the meaning of indicators.

Author Response

Legend used:

Green: our reply

Black: reviewer comment

Blue: what was added to text

Reviewer 2:

General comments:

Vitamin D-related disorders may be caused by polymorphic variants of genes coding protein molecules involved in vitamin D metabolism and transport. However, no published bibliometric analysis covered any vitamin D metabolism gene polymorphisms (vitDMGPs). This manuscript presents a bibliometric investigation for vitDMGPs. Although this paper is interesting, I have several concerns:

Thank you, we are happy that you liked our paper.

Thank you so much for you for your feedback, we appreciate your efforts to improve our manuscript. Please see our response to each points below.

We are very grateful to the reviewer for their patience, careful reading of our manuscript and for providing us with excellent guidance to increase the accuracy of our results and enhance the paper's value. We have followed your advice and added a reply point-by-point. We will also be very grateful if you have further comments to enhance our paper.

  1. This paper uses bibliometric investigation to study vitDMGPs, which has certain research significance, but it is not appropriate to say that novel insights, and this paper has a limited role in promoting the diagnosis and treatment of a spectrum of diseases. Thus, The consistency of the text topic needs to be strengthened, such as“Research trends of vitamin D metabolism gene polymorphisms based on a bibliometric investigation”.

Thank you, we agree with you, and we changed the title as suggested to:

“Research trends of vitamin D metabolism gene polymorphisms based on a bibliometric investigation.”

  1. The time range of literature research needs to be checked. In the abstract, the research period is from 1998 to 2022, but in the Database and search methodology section, the research period is set as 1950 to 2022.

Kindly note that 1950 to 2022 was the original research query, then we find only literature start from 1998 which was all represented in Fig 1a.

  1. In bibliometric investigation, the depth and breadth of literature research are very important, but I noticed that the scope of research in this paper is limited to articles in English, and the author should expand the scope of research in papers in other languages. In addition, processing papers, letters and early access articles may usually include the latest research results in this field, which is inappropriate to delete.

Thank you very much for this great comment which highlights your experience in this field. And kindly allow us to address this question. First of all, we totally agree with you that in-depth analysis is required; hence, all languages are required. We performed this technique in our previous paper, [24] in our manuscript, when we targeted only one gene, which was the VDR:

[24]      Abouzid, M.; Główka, A.K.; Karaźniewicz-Łada, M. Trend Research of Vitamin D Receptor: Bibliometric Analysis. Health Informatics J. 2021, 27, 14604582211043158, doi:10.1177/14604582211043158.

In this paper, we performed various trials of research query and inspected all other languages and types, and interestingly that did not show up any new information in the case of trending topics or top 10 or top 25 citation bursts.

For example, there are only 23 articles in languages other than English. The most cited paper was cited 28 times, and it was a case series on Smith-Lemli-Opitz Syndrome, which was already identified in our paper in a separate cluster.

Also, regarding why we selected particular articles, please note that we needed to highlight what is already proven, for instance, selecting review articles will result in un-refined data because they will be more repetition of the several keywords. But we indeed analyze the reviews indirectly when we perform citation bursts that highlight for us the most important review articles in the entire field, and that was achieved in section 4.2.2.

Still, we trust that processing papers, letters and early access articles may usually include the latest research; however, since they are new and our analysis is setting a minimum repletion of a keyword to at least 5 times (and 5 genes), it still not appears in the trending topics. Nevertheless, we have also added this part to our limitation so future scholars can consider, and now it reads:

We included only published articles, and the exclusion criteria for analyzing trending keywords were set to a minimum of five. Thus, at the time of writing this paper, newly published remarkable documents (processing papers, letters, and early access articles) were excluded from the trend analysis.

  1. In Section 3.7, the article uses some indicators to display the research results, such as Modularity Q, late semantic indexing (LSI), Log like food ratio (LLR), and mutual information (MI), but does not clearly explain the meaning of indicators.

Thank you, we have explained these indicators and now the section reads as below (what added is marked blue).

  1. Lines 225 to 232

“The timeline view of clustered maps of co-cited 2,321 references from 2005 to 2022 yielded 19 clusters according to keywords. According to Chen "the modularity of a network measures the extent to which a network can be decomposed to multiple components, or modules” [49, page 82]. Therefore, modularity Q of 0.7696 is relatively high, so the network is reasonably divided into loosely coupled clusters. Silhouette value can be used as an indicator for the quality of clustering configuration and it ranges between -1 and 1 [49]. The mean silhouette score of 0.8909 suggests relatively high homogeneity of the clusters [49].”

  1. Following explanation was add as table 2 footer.

α LSI is based on a singular value decomposition of the term by document matrix [69]

β LLR uses the asymptotic distribution of the generalized likelihood ratio, which improves the statistical results. For textual analysis, LLR can be applied to smaller text volumes and allow the comparison between the significance of the occurrence of both rare and common phenomenon [70]

γ MI measures the degree of relatedness between two variables [71]

  1. Lines 245-249 changes:

Clusters labels are shown in Table 1 according to different labels: latent semantic indexing (LSI), Log-likelihood ratio (LLR), and mutual information (MI) for better clarity of the dynamicity of the publications on the timeline (Table 2). LSI shows the most important aspect of a cluster, whereas LSI and MI tend to reflect a unique aspect of it.

Round 2

Reviewer 2 Report

The authors have basically addressed the points raised by this Reviewer in the revised version, which is now in good shape for publication.
